# USP7 Induces Chemoresistance in Triple-Negative Breast Cancer via Deubiquitination and Stabilization of ABCB1

**DOI:** 10.3390/cells11203294

**Published:** 2022-10-19

**Authors:** Yueh-Te Lin, Joseph Lin, Yi-En Liu, Yun-Cen Chen, Shiang-Ting Liu, Kai-Wen Hsu, Dar-Ren Chen, Han-Tsang Wu

**Affiliations:** 1Cancer Genome Research Center, Department of Medical Research and Development, Chang Gung Memorial Hospital at Linkou, Guishan District, Taoyuan 333, Taiwan; 2Comprehensive Breast Cancer Center, Changhua Christian Hospital, Changhua 500, Taiwan; 3Cancer Research Center, Changhua Christian Hospital, Changhua 500, Taiwan; 4Department of Nursing, Changhua Christian Hospital, Changhua 500, Taiwan; 5Research Center for Cancer Biology, Institute of New Drug Development, China Medical University, Taichung 404, Taiwan; 6School of Medicine, Chung Shan Medical University, Taichung, 402, Taiwan

**Keywords:** ubiquitin specific protease 7, chemoresistance, triple-negative breast cancer, ATP-binding cassette B1

## Abstract

Triple-negative breast cancer (TNBC) accounts for 15–20% of all breast cancer. TNBC does not express the estrogen receptor, progesterone receptor, or human epidermal growth factor receptor 2. Cytotoxic chemotherapy and surgery are the current therapeutic strategies for TNBC patients, but the chemoresistance of TNBC limits the efficiency of this strategy and shortens the lifespan of patients. The exploration of targeted therapy is ongoing in TNBC research. The aim of the present study was to identify the mechanism underlying acquired resistance in TNBC through the exploration of the relationship between the expression of USP7 and of ABCB1. We found that ubiquitin specific protease 7 (USP7) is a potential therapeutic target for overcoming the chemoresistance of TNBC. USP7 overexpression increased the chemoresistance of TNBC, while the knockdown of USP7 effectively increased the chemosensitivity of chemoresistant TNBC. A USP7 inhibitor effectively induced apoptosis and suppressed metastasis in chemoresistant TNBC. We further clarified that USP7 is a specific deubiquitinating enzyme for ABCB1 that plays an essential role in drug resistance. USP7 directly interacted with ABCB1 and regulated its stability. We concluded that USP7 promotes the chemoresistance of TNBC by stabilizing the ABCB1 protein.

## 1. Introduction

Breast cancer (BC) is the most commonly diagnosed malignancy globally. In 2020, an estimated 2.26 million cases were diagnosed, with 685,000 deaths resulting from the disease, according to the World Health Organization [1]. BC is a heterogeneous disease, and is currently divided into three major subtypes based on gene expression profiles: estrogen receptor (ER) positive (luminal A and B), human epidermal growth factor receptor (HER2) positive, and triple-negative BC (TNBC) [2,3]. Different biological subtypes are further associated with discrepancies in treatment responses and disease-specific outcomes [4,5]. The outcome for patients diagnosed with HER2 enriched or ER-positive BCs has been improved by the development of targeted therapies which significantly reduce the risk of death and relapse. However, in patients with TNBC, treatment options remained limited, and the prognosis is relatively poor. TNBC tumors are resistant to endocrine therapy and HER2-targeting treatments, making cytotoxic chemotherapy the only approved and validated treatment in adjuvant and neoadjuvant settings [6].

Although optimal protocols are yet to be determined, anthracycline and taxane-based regimens are presently the backbone of systemic therapeutics for TNBC. A platinum-based regimen has produced favorable results in neoadjuvant and metastatic settings [7]. However, the development of chemoresistance often limits the efficacy of current standard therapeutic regimens, and the establishment of an optimal treatment for TNBC remains the biggest challenge in increasing the overall survival rate in this group of patients. Many mechanisms which may contribute to the development of anticancer drug resistance have been studied [8]. Of these, ATP-binding cassette (ABC) transporter-mediated drug efflux has received considerable attention [9,10]. The overexpression of ABC transporters is related to both anthracycline and taxane resistance in breast cancer, since doxorubicin and paclitaxel are substrates of p-glycoprotein (Pgp) encoded by the ABCB1 gene [11,12,13,14]. Thus, a rational approach to overcoming chemoresistance is to target the ABCB1 transporter using activity inhibitors. A wide range of ABCB1 transporter inhibitors have been studied, but they are either too toxic to be beneficial, lack specificity, or have little impact on drug accumulation [15,16,17]. Regardless of these poor initial results, research into ABCB1 transporter inhibitors is ongoing. These findings suggest that the development of TNBC chemoresistance may be related to multiple pathways, and further exploration of the mechanisms that directly induce ABCB1 overexpression is warranted.

Ubiquitination is a reversible, post-translational modification that covalently conjugates a ubiquitin (Ub) moiety onto a substrate molecule, a process which may further regulate the activity, interactions, and stability of substrate proteins [18]. Ubiquitination of protein is constantly reversed by deubiquitinating enzymes (DUBs) that remove Ub from substrates and therefore keep them from proteosome-dependent degradation [19]. Ubiquitin specific protease (USP) 7 is one of the most widely studied DUBs functioning in proteosome-dependent degradation [20]. A large number of studies have demonstrated an association between USP7 expression and the development of drug resistance in multiple malignancies, in response to treatment with chemotherapeutic agents [21]. Raised USP7 expression may suggest a poor tumor prognosis; therefore, it has been considered by some researchers to be a marker for prognosis and a potential drug target for anticancer therapy [22].

Dysfunction of the ubiquitination and proteasomal system, especially the degradation of ABCB1, plays a crucial role in anticancer drug resistance [23,24]. These results led us to investigate the regulation of the stability of ABCB1. Ubiquitination modification functions are not limited to protein degradation, but also include the regulation of other biological processes, including DNA replication and repair, receptor response, protein trafficking, gene transcription, and protein-protein interactions [25]. In the present study, we thought to identify the mechanism underlying acquired resistance in TNBC through the exploration of the relationship between the expression of USP7 and of ABCB1. We evaluated the expression and effects of USP7 in chemoresistant TNBC cells, and showed that ABCB1 expression is directly regulated by USP7, a member of the DUBs, which in turn regulates the chemoresistance of TNBC through the USP7-ABCB1 pathway.

## 2. Material and Method

### 2.1. Chemicals

Doxorubicin (98% of purity) and paclitaxel (98% of purity) were obtained from Cayman Chemical (East Ellsworth Rd., MI, USA). GNE-6776 (95% of purity) was obtained from Carbosynth (Old Station Business Park, Compton, UK). The stock solution of GNE 6776 was dissolved in dimethyl sulfoxide (DMSO) (Sigma, St. Louis, MO, USA) and stored at −20 °C. The less than 0.1% of DMSO in culture medium was used in all treatments.

### 2.2. Cell Culture

MDA-MB-231 cells was purchased from the ATCC (American Type Culture Collection). MDA-MB-231 cells was treated with doxorubicin or paclitaxel, and selected resistance cells, MDA-MB-231-DoxR and MDA-MB-231-PtxR, respectively. MDA-MB-231-DoxR cells were incubated in DMEM (Dulbecco’s modified Minimal Essential Medium) medium with high glucose, and 10% of FBS containing 5 nM of doxorubicin and MDA-MB-231-PtxR cells was incubated in DMEM (Dulbecco’s modified Minimal Essential Medium) medium with high glucose and 10% of FBS containing 2 nM of paclitaxel. The cell lines tested negative for mycoplasma contamination. 

### 2.3. Plasmids

The HA-ABCB1 plasmids were obtained from Addgene (plasmid #10957). The various plasmids were shown in Appendix A. 

### 2.4. Cell Viability/Cytotoxicity Assay

MDA-MB-231 were transfected different USPs (7, 12, 21), or knockdown of USP7, respectively. The cells were seeded in 96-well plates and treated with various concentrations of doxorubicin (5, 7.5, 10, 12.5 nM) or paclitaxel (2, 4, 8, 10 nM) for 72–96 h. For the effect of GNE-6776 on MDA-MB-231, MDA-MB-231-DoxR and MDA-MB-231-PtxR cell lines. Cells were seeded in 96-well plates, treated with GEN-6776 at different concentrations (5, 10, and 15 μM) for 72–120 h. The 3-(4,5-dimethylthiazol-2-yl)-2,5-diphenyltetrazolium bromide (MTT) assay was performed for detection of cells viability and described before [20,21]. The absorbance was measured at 595 nm using an ELISA plate reader. 

### 2.5. Colony Formation Assay

The cells were seeded in 12-well plates for 2–3 weeks later, fixed in methanol at room temperature for 20 min, and then stained with 0.5% crystal violet (Sigma, St. Louis, MO, USA) for 35 min. Images were obtained using a digital camera. The crystal violet was dissolved in 33% acetic acid. Absorption was measured at 550 nm using an ELISA plate reader. 

### 2.6. Cell Proliferation Assay

The cells were seeded in 96-well plates and treated with different concentrations of doxorubicin (5, 7.5, 10, 12.5 nM), Paclitaxel (2, 4, 8, 10 nM) for 0, 24, 48, 72 h. The process in detail was described before [26].

### 2.7. Apoptosis Analysis

MDA-MB-231 were transfected different USPs (7, 12, 21) or knockdown of USP7, respectively. Cells were followed by plated in 96-well overnight and treated with different concentrations of doxorubicin (5, 7.5, 10, 12.4 nM) or paclitaxel (2, 4, 8, 10 nM) for 96 h. On the other hand, MDA-MB-231-DoxR and MDA-MB-231-PtxR cells were plated in 12-well plates overnight, and treated with various concentrations of GNE-6776 for 96 h. The harvested cells were incubated in Muse^TM^ Annexin-V and Dead Cell kit reagent (Merck Millipore, Billerica, MA, USA) at room temperament for 20 min in the dark, and analyzed by Muse^TM^ Cell Analyzer (Merck Millipore, Billerica, MA, USA). The data were analyzed using MUSE 1.7 Analysis software (Merck Millipore, Billerica, MA, USA).

### 2.8. Mitochondrial Membrane Potential Analysis

Depolarized levels of Mitochondrial Membrane Potential for cells were determined using Muse^TM^ Cell Analyzer (Merck Millipore, Billerica, MA, USA). Briefly, cells were seeded in 12-well plates overnight and treated with different concentrations of doxorubicin, paclitaxel, or GNE-6776 for 72 h, respectively. The harvested cells were resuspended in Muse^TM^ MitoPotential Kit reagent (Merck Millipore, Billerica, MA, USA), and according to the manufacturer’s protocol. The data were analyzed using MUSE 1.7 Analysis software (Merck Millipore, Billerica, MA, USA).

### 2.9. Migration and Invasion Assays

Falcon^®^ Cell Culture Inserts (CORNING) and BD BioCoat Matrigel Invasion Chambers (Becton Dickson) were used for migration (2 × 10^4^ cells/well) and invasion (2 × 10^5^ cells/well containing 12 μL matrigel with growth factor, BD) assays, which were performed using 24-well plates for 18 and 24 h, respectively, as described previously. GNE-6776 (50 mg/mL in stock) was used to treat cells for 12 h, and then plates were replaced with fresh medium. The scale bar represents 200 μm.

### 2.10. In-Vivo Xenograft Studies

All animal work was carried out in accordance with The AAALAC international for animal care. Animal experiment in this study was followed by 3Rs guideline and approved by Changhua Christian Hospital. The approval number: CCH-AE-107-010 and the data of approval is 21st of December, 2018.

Seven- to eight-week-old female SCID mice (NLAC, Taiwan) were housed under specific pathogen-free conditions and provided with food and water ad libitum. As described previously [20,22], we numbered mammary ducts 1–10 in the female mice. On day 0, orthotopic mammary tumors were inoculated with the breast tumor cell line MDA-MB-231-DoxR and MDA-MB-231-PaxR (10 × 10^7^ cells in saline) at the mammary duct. Tumors were measured twice a week with digital microcalipers, and volumes were calculated (volume = (width^2^ × length)/2). Tumors were measured twice weekly, and once the maximal tumor volume was reached (600 to 800 mm^3^), mice were sacrificed. Tumor volumes are represented as mean volume ± s.d. The number of observed nodules in the lungs was calculated and represented as mean volume ± s.d. The scale bar respresents 1 cm.

### 2.11. Protein Extraction, Western Blot Analysis, RNA Extraction, and Quantitative Real-Time PCR

The extraction of proteins from cells and Western blot analysis was performed as described [27]. The characteristics of the antibodies used were listed (Appendix A). RNA purification, cDNA synthesis, and quantitative real-time PCR analysis were performed as described before [6]. The sequences of primers used in the real-time PCR experiment were shown (Appendix A).

### 2.12. Co-Immunoprecipitation and GST Pull-Down Assays 

Co-immunoprecipitation experiments were performed by incubating different antibodies (Appendix A) and were described before [23]. GST pull-down assays were performed by incubating HA-CDK1 with GST-USP7 1–209 or GST-USP7 210–500 protein and Glutathione-Agarose (Sigma, St. Louis, MO, USA). Purification of GST proteins was described [22]. The pulled-down HA-CDK1 protein was detected by Western blot analysis. 

### 2.13. Deubiquitination Assays

For the in vivo deubiquitination assay, knockdown of USP7 silencing in MDA-MB-231 cells was performed by lentivirus infection system with shRNA. MDA-MB-231-USP7-silencing cells were treated with 50 mM of MG132 (Sigma, St. Louis, MO, USA) to prevent CDK1 protein degradation for 12–16 h and lysed by RIPA buffer. For the in vitro deubiquitination assay, Flag-USP7 wild type and its mutants were expressed in 293T cells, lysed by RIPA buffer, immunoprecipitated by anti-Flag agarose (Sigma, St. Louis, MO, USA), and eluted by Flag peptides (100 mg/mL) in the DUB buffer (50 mM Tris-HCl pH 8.0, 50 mM NaCl, 1 mM EDTA, 10 mM DTT, 5% glycerol). The Myc-ubiquitinated HA-CDK1 were expressed in 293T cells with MG132 treatment, precipitated by protein A beads conjugating with anti-HA antibodies (Sigma, St. Louis, MO, USA), and eluted by 100 mM of HA peptide (Sigma, St. Louis, MO, USA) in the DUB buffer. The process in detail was described before [23]. 

### 2.14. Statistical Analysis

Data are represented as the mean ± SEM for biological triplicate experiments. All error bar analysis was by STDEV. One-way ANOVA was used to compare different groups at the same concentration; a *p* value of <0.05 was considered statistically significant.

## 3. Results

### 3.1. USP7 Specifically Promotes the Chemoresistance of TNBC

Three USP family members, USP7, USP12, and USP21, were first tested for their role in the chemoresistance of TNBC. USP7, USP12, and USP21 were transiently expressed in MDA-MB-231 and MDA-MB-468 cells, and were then treated with a serial does of doxorubicin (Dox) or paclitaxel (Ptx). The 3-(4,5-Dimethylthiazol-2-yl)-2,5-Diphenyltetrazolium Bromide (MTT) experiment indicated that USP7 specifically increases the chemo-drug tolerance of TNBC cells in response to either doxorubicin or paclitaxel treatment (Figure 1A,B and Appendix A). Colony formation assays showed that the growth of MDA-MB-231/ MDA-MB-468-USP7-expressing cells was significantly higher than that of MDA-MB-231-expressing empty vector (Ctrl) under increasing doses of Dox or Ptx (Figure 1C,D and Appendix A). The apoptotic activity of MDA-MB-231 and MDA-MB-468 cells expressing USP7, USP12, or USP21, treated with increasing doses of Dox or Ptx was further assessed using Annexin V-dependent flow cytometry analysis. USP7 specifically repressed Dox or Ptx-induced apoptosis in TNBC cells (Figure 1E,F and Appendix A). Moreover, only USP7 expressions are consistently higher in TNBC cell lines compared with the normal breast cell line (Appendix A). 

### 3.2. USP7 Regulates Drug Resistant Activity in Chemoresistant TNBC

To clarify the role of USP7 in the chemoresistance of TNBC, doxorubicin-resistant MDA-MB-231 and MDA-MB-468 cells with doxorubicin-resistance (MDA-MB-231-DoxR, MDA-MB-468-DoxR), paclitaxel-resistant MDA-MB-231 and MDA-MB-468 cells with paclitaxel resistance (MDA-MB-231-PtxR, MDA-MB-468-PtxR) were generated using a step-by-step increasing dose treatment. Doxorubicin-resistant and paclitaxel-resistant TNBC cells had higher cell viability and growth than parental MDA-MB-231 cells under Dox or Ptx treatment, respectively (Figure 2A–C and Appendix A). MDA-MB-231-DoxR-USP7-silencing and MDA-MB-231-PtxR-USP7-silencing cells were generated using a lentivirus infection system carrying USP7 shRNA, and used to further investigate the relationship between USP7 and chemoresistance in TNBC. The knockdown of USP7 repressed the viability of MDA-MB-231-DoxR and MDA-MB-231-PtxR cells under serial doses of Dox and Ptx treatments, respectively (Figure 3A–C). Lower proliferative activity of MDA-MB-231-DoxR-USP7-silencing and MDA-MB-231-PtxR-USP7-silencing cells was observed at 5 nM Dox and 2 nM Ptx, respectively, for 24–72 h (Figure 3D). The repression of USP7 also significantly induced apoptosis in MDA-MB-231-DoxR and MDA-MB-231-PtxR cells under serial doses of Dox and Ptx (Figure 3E,F). The expression levels of the apoptosis-related proteins cleaved PARP, caspase 3, and caspase 7 were markedly increased in MDA-MB-231-DoxR-USP7-silencing and MDA-MB-231-PtxR-USP7-silencing cells at 10 nM of dox and 8 nM of Ptx treatment, respectively (Figure 3G). Since apoptosis is related to disruption of the mitochondrial membrane potential (MMP), we further investigated the relationship between USP7 and MMP in MDA-MB-231-DoxR and MDA-MB-231-PtxR cells. The knockdown of USP7 depolarized the MMP in MDA-MB-231-DoxR and PtxR cells at serial doses of Dox and Ptx in a dose-dependent manner (Figure 3H,I). The MMP-related proteins Bcl2 and BclxL were markedly decreased; BAX and Bim were highly increased in MDA-MB-231-DoxR and PtxR-USP7-silencing cells under 10 nM of Dox and 8 nM of Ptx treatment, respectively (Figure 3J). The similar results were also observed in MDA-MB-468-DoxR and MDA-MB-468-PtxR cells with USP7 suppression (Appendix A). 

### 3.3. USP7 Enhances the Metastatic Activity in Chemoresistant TNBC

Since drug resistance causes cancer cells to exhibit increased proliferation and aggression, we further assessed the metastatic ability of chemoresistant TNBC with USP7 suppressed. The migration ability of MDA-MB-231-DoxR-USP7-silencing and MDA-MB-231-PtxR-USP7-silencing cells was repressed compared with control, according to in vitro migration assays (Figure 4A,B). The decreased invasive activity of MDA-MB-231-DoxR-USP7-silencing and MDA-MB-231-PtxR-USP7-silencing cells was assessed using in vitro invasion assays (Figure 4C,D). The markers for epithelial–mesenchymal transition (EMT) were evaluated in MDA-MB-231-DoxR-USP7-silencing and MDA-MB-231-PtxR-USP7-silencing cells using western blotting analysis. Increased levels of E-cadherin and plakoglobin, and decreased levels of vimentin and N-cadherin were observed compared with control (Figure 4E). The viability of MDA-MB-231-DoxR and MDA-MB-231-PtxR cells was decreased by knockdown of USP7, but re-expression of USP7 increased the viability of MDA-MB-231-DoxR-USP7-silencing and MDA-MB-231PtxR-USP7-silencing cells under Dox and Ptx treatment (Figure 5A,B). The migration and invasive activities of MDA-MB-231-DoxR and MDA-MB-231-PtxR cells were decreased by knockdown of USP7, but re-expression of USP7 and ABCB1 increased the metastatic activity of MDA-MB-231-DoxR and MDA-MB-231-PtxR-USP7-silencing cells under Dox and Ptx treatment (Figure 5C–F). The viability, migration, and invasive activities of MDA-MB-468-DoxR and MDA-MB-468-PtxR cells were also regulated by UPS7 (Appendix A).

### 3.4. Inhibition of USP7 Induces Apoptosis of Chemoresistant TNBC

USP7 inhibitor (GNE-6776) was used to further clarify the role of USP7 in the chemoresistance of TNBC. Treatment of MDA-MB-231-DoxR/-PtxR cells with GNE-6776 alone or combined with Dox or Ptx was evaluated using MTT assays. All treatments significantly decreased the viability of MDA-MB-231-DoxR and MDA-MB-231-PtxR cells (Figure 6A,B). Serial dose treatment with GNE-6776 also significantly increased apoptosis in MDA-MB-231-DoxR and MDA-MB-231-PtxR cells (Figure 6C,D). The apoptosis-related proteins cleaved PARP, caspase-3, 7, 8, and 9 were increased in MDA-MB-231-DoxR and MDA-MB-231-PtxR cells treated with serial doses of GNE-6776 (Figure 6E). Depolarization of the MMP was also observed in MDA-MB-231-DoxR and MDA-MB-231-PtxR cells treated with GNE-6776, in a dose-dependent manner (Figure 6F,G). Decreased protein levels of the Bcl family members Bcl-2 and Bcl-xL and increased expression of Bim and BAX were observed in GNE-6776-treated MDA-MB-231-DoxR and MDA-MB-231-PtxR cells (Figure 6H). GNE-6776 also induce apoptosis in MDA-MB-468-DoxR and MDA-MB-468-PtxR cells (Appendix A).

### 3.5. USP7 Inhibitor Suppresses Metastasis of Chemoresistant TNBC

The migration ability of MDA-MB-231-DoxR and MDA-MB-231-PtxR cells was repressed by GNE-6776 in a dose-dependent manner (Figure 7A,B). A serial dose of GNE-6776 treatment cause decreased invasive activity of MDA-MB-231-DoxR and MDA-MB-231-PtxR cells in a dose-dependent manner (Figure 7C,D). The expression of epithelial markers, E-cadherin, and plakoglobin was increased, and those of mesenchymal markers, vimentin, and N-cadherin were decreased in MDA-MB-231-DoxR and MDA-MB-231-PtxR cells treated with serial doses of GNE-6776 (Figure 7E). Repression of the migration and invasive activities in MDA-MB-468-DoxR and MDA-MB-468-PtxR cells by GNE-6776 were also observed (Appendix A).

### 3.6. USP7 Regulates ABCB1 Expression in TNBC

We further explored downstream candidates regulated by USP7 involved in the chemoresistance of TNBC. The ABC family members ABCB1, ABCC1, and ABCG2, which are related to drug resistance in several types of cancer, were first evaluated in MDA-MB-231 cells. Only ABCB1 expression was significantly downregulated in MDA-MB-231-USP7-silencing cells compared with control (Figure 8A). The mRNA level of ABCB1 was not changed in MDA-MB-231-USP7-silencing cells compared with control (Figure 8B). Re-expression of USP7 rescues the protein level of ABCB1 in MDA-MB-231-USP7-silencing cells (Figure 8C). The expression of ABCB1 in different TNBC cell lines (MDA-MB-231, BT20, and BT549) was higher than its levels in ER-positive (ER+) BC cell lines (Figure 8D). Higher expression of ABCB1 was observed in MDA-MB-231-DoxR and MDA-MB-231-PtxR cells than in parental cells (Figure 8E). The expression of ABCB1 was repressed in MDA-MB-231-DoxR-USP7-silencing and MDA-MB-231-PtxR-USP7-silencing cells (Figure 8F). USP7 upregulated ABCB1 expression in a dose-dependent manner in 293T cells by transient transfection assay (Figure 8G). Re-expression of ABCB1 in MDA-MB-231-DoxR/PtxR cells promoted chemoresistance activity in response to Dox and Ptx, respectively (Figure 8H,I).

### 3.7. USP7 Is a Specific Deubiquitinating Enzyme for ABCB1

USP7 is a deubiquitinating enzyme that removes the subtract from the K48-linked polyubiquitin chain and thus stabilizes its protein expression. We first performed co-immunoprecipitation (CoIP) assay experiments to detect interactions between USP7 and ABCB1. ABCB1 was observed in MDA-MB-231-USP7-silencing cells by pulling down the USP7 with ant-USP7 antibodies (Figure 9A). Flag-tag USP7 interacts with ABCB1 in 293T cells with transient expression of Flag-tag USP7, according to immunoprecipitation experiments using anti-Flag antibodies (Figure 9B). The N-terminal domain (1–500 aa) of USP7 is the domain which interacted with ABCB1 (Figure 9C). The reverse experiment also showed the same result (Figure 9D). USP7 directly interacts with ABCB1 according to GST-pull down assays, and the USP7 domain directly interacting with ABCB1 is 1–209 aa (Figure 9E). We previously identified that 1–209 aa of USP7 is the protein-protein interaction region [27]. In vivo deubiquitination assays were performed to detect whether ABCB1 is a substrate for USP7 by treating MDA-MB-231-USP7-silencing cells with MG132 and then immunoprecipitating with endogenous ABCB1. Knockdown of USP7 increased the K48-linked polyubiquitin chain of ABCB1 compared with control (Figure 9F). We further evaluated whether USP7 is a deubiquitinating enzyme for ABCB1 by performing in vitro deubiquitination assays. Purified flag-tag USP7 wild-type (WT) removed K48-linked polyubiquitin chains from purified polyUb-HA-ABCB1. The level of K48-linked polyubiquitin chains of ABCB1 incubated with a loss of function mutant (K443R) of USP7 [27] was similar to that of ABCB1 without USP7 (Figure 9G).

### 3.8. USP7-Silencing Chemoresistant TNBC Exhibits Significantly Reduced Tumorigenesis and Lung Metastasis in Orthotopic BC Mouse Models

We orthotopically injected the MDA-MB-231-DoxR and MDA-MB-231-PtxR cell lines into female NOD SCID mice. On Day 16, the tumor volumes of mice orthotopically injected with USP7-silencing cell lines mice began to decline in comparison with control mice (Figure 10A,B), even though the tumors of MDA-MB-231-DoxR and MDA-MB-231-PtxR xenograft models had different growth rates. The tumor disappeared in one MDA-MB-231-PtxR USP7-silencing xenograft model. USP7-silencing cell lines produced significantly decreased numbers of nodules in mice lungs compared to the control group (Figure 10C,D). Taken together, these results suggested that USP7 plays an important role in tumorigenesis and metastasis in BC xenograft mouse models.

## 4. Discussion

TNBC accounts for 10–20% of annually diagnosed BC, cases and often occurs in younger females [28]. The absence of ER, PR expression, and HER2 amplification currently makes chemotherapy the most effective medical treatment [6]. Although optimal protocols are yet to be determined, taxane and anthracycline-based regimens are presently the backbone of systemic therapeutic options for TNBC. Platinum-based regimens have produced favorable results in neoadjuvant and metastatic settings [7]. Despite having a worse prognosis, patients with TNBC tumors have better pathological complete response (pCR) rates than non-TNBC patients after neoadjuvant chemotherapy [29,30]. However, Liedtke et al. demonstrated that TNBC patients with residual disease are more likely to suffer from early recurrence and die from metastatic disease [29]. The discrepancies between high sensitivity to neoadjuvant therapy and poor clinical prognosis driven by higher relapse rates among patients with residual disease suggests that a considerable proportion of TNBC tumors become drug resistant during treatment, or are inherently resistant.

USP7 was initially believed to function as a p53 deubiquitinating enzyme, resulting in the stabilization of p53 [31,32]. Consistent with this assessment, one study also showed a correlation between low USP7 expression levels and poor patient prognosis in non-small cell lung cancer [33]. However, USP7 was demonstrated to have an oncogenic function in BC, acting by regulating the stability of geminin, an inhibitor of the DNA replication licensing factor, whose levels were shown to be prognostic for poor BC survival in another study [34]. USP7 was shown to deubiquitinate and stabilize the ERα subunit, promoting cell proliferation and tumor growth in ER-positive BC by inhibiting cell cycle arrest and apoptosis [35]. These findings suggest that overexpression of USP7 is an indicator of poor prognosis for BC patients. Upregulation of USP7 is frequently observed in various types of cancer, and the apparent contribution of this upregulation to carcinogenesis has led to speculation that USP7 could be an effective target in anticancer therapies.

In the current study, we utilized established cell lines in which MDA-MB-231 BC cells were selected for survival in increasing concentration of doxorubicin (MDA-MB-231-DoxR) or paclitaxel (MDA-MB-231-PtxR), to better understand the mechanism underlying acquired resistance in TNBC. Cell lines selected under identical conditions with USP7 knockdown were used for comparison. We found that USP7 expression promoted and regulated the chemoresistance of TNBC cells. Silencing of USP7 expression decreased the proliferative and metastatic ability of chemoresistant TNBC cells. A USP7 inhibitor exerted anticancer activity by inducing apoptosis and repressing metastasis of resistant TNBC. We further found that TNBC chemoresistance is regulated by the USP7-ABCB1 pathway. A number of scientific reports have proposed that USP7 is a key effector protein in the emergence of resistance to doxorubicin and paclitaxel chemotherapies in various cancers [20,36,37]. We discovered for the first time that USP7 is a novel regulator of ABCB1 activity and expression. Therefore, USP7 may contribute to resistance via its role in stabilizing ABCB1, which is responsible for the apical transport of these two agents. We revealed that modulation of USP7 expression levels and treatment with USP7 inhibitor led to the downregulation of ABCB1 through the USP7-ABCB1 pathway in anthracycline and taxane-resistant TNBC cells. Besides, PIK3CA (p110α), markers for cell cycle (CDK6, cyclin B1) and proliferation (Ki67, PCNA) were downregulated in chemoresistant TNBC cells by USP7 inhibitor (Appendix A)

Other mechanisms which may contribute to the development of anticancer drug resistance have also been studied [8]. Among them, ABC transporter-mediated drug efflux has received considerable attention [9,10]. Forty-nine different ABC transporters have been identified in humans, classified into seven subfamilies, A–G [38]. ABCB1 is the most well-studied member of the family, and a large number of studies have demonstrated that ABCB1 expression increases after chemotherapy [39,40]. ABCB1 expression was shown to be higher and more frequently observed in TNBC patients than in ER-positive subtypes, even before the administration of cytotoxic agents [41]. This observation suggests that this transporter may also provide an intrinsic growth advantage to cancer cells and promote their aggressive nature.

Our results support those of many previous studies which have found an ABCB1-mediated resistance mechanism in doxorubicin- and paclitaxel-resistant BC cell lines [11,12]. This mechanism is of considerable importance in TNBC, as doxorubicin (anthracycline) and paclitaxel (taxane), which are used routinely in first-line chemotherapy protocols, are well-studied substrates of the ABCB1 transporter [11,12]. Doxorubicin is a cell cycle agent that acts directly on DNA to interfere with DNA transcription, therefore preventing mRNA synthesis and causing cell death. Paclitaxel belongs to a class of cytotoxic drugs, taxanes, that act on the G2 and M phases of the cell cycle to promote the irreversible accumulation of tubulin, resulting in abnormal chromosomal partitioning during the formation of the mitotic spindle. This phenomenon prevents proper mitotic cell division and eventually leads to cell death [18]. Because cancer cells with drug transporters are responsible for resistance to chemotherapy agents, earlier trials have attempted to combine chemotherapeutic drugs with newer target agents to understand the mechanisms underlying tumor invasion, that might reduce TNBC-associated mortality rates [42]. However, the addition of the ABCB1 inhibitor has not succeeded in restoring sensitivity to chemotherapy or improving clinical outcomes in BC patients [15,16,17]. Therefore, most trials were terminated due to severe side effects, and no health authority-approved inhibitors of ABC transporter currently exist [43]. These data also suggested the development of TNBC chemoresistance may be related to the number and diversity of pathways involved, and further exploration of mechanisms that directly induce ABCB1 overexpression is warranted. In future directions, another prominent approach to targeting these transporters to overcome chemoresistance is by inhibiting their expression, rather than using an ABCB1 inhibitor to suppress their activity. We found that modification of USP7 expression might be a valuable tactic to overcome the limitations of anthracycline and taxane resistance. Our results demonstrated pharmacological inhibition of USP7, which further downregulated ABCB1 expression, suppressed tumor growth, and considerably restored the effectiveness of doxorubicin and paclitaxel. We further used the mouse xenograft model to identify the regulatory role of USP7 in TNBC chemoresistance in vivo. The results clearly demonstrated that xenograft tumors from USP7-silencing cell lines were significantly smaller than those of controlled mice. Moreover, silencing of USP7 also markedly inhibited lung tumor metastases in mouse models. Another future focus is how these USP7 inhibitors should be implemented, whether as a single agent or combined treatment. Due to the extensive heterogeneity of TNBC and existence of crosstalk between signaling pathways, it will be interesting to learn whether and how different mechanisms regulate the TNBC chemoresistance independently or coordinately in the future study. Going forward, the requirement of incorporating and validating biomarkers in the future clinical studies of TNBC is also evident.

## 5. Conclusions

Collectively, our data produced previously unreported evidence that both doxorubicin- and paclitaxel-induced chemoresistance could be reversible following combination treatment with a USP7 inhibitor. In addition, we report for the first time that ABCB1 is a novel substrate of USP7, and USP7 maintains the protein stability of ABCB1. We revealed that modulation of USP7 expression levels and treatment with USP7 inhibitor led to the downregulation of ABCB1 via the USP7-ABCB1 pathway in anthracycline and taxane-resistant TNBC cells. Our findings provide preclinical evidence that USP7 is an attractive target for treatment of drug resistant TNBC, acting via regulation of the USP7-ABCB1 pathway, and offer a rationale for further assessment of USP7 inhibitor for the treatment of TNBC.

## Figures and Tables

**Figure 1 cells-11-03294-f001:**
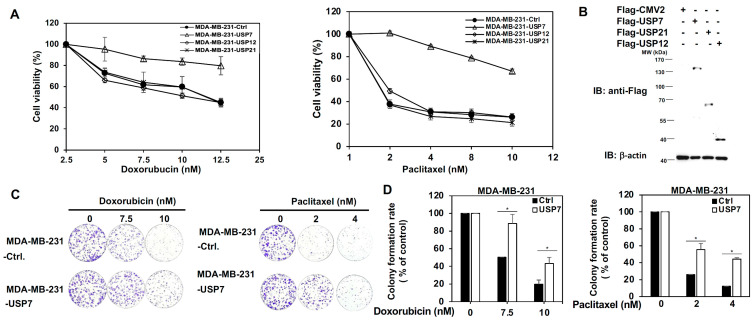
Role of ubiquitin specific protease (USP) 7 on the acquired chemoresistance of triple-negative breast cancer (TNBC). (**A**) Analysis of the viability of MDA-MB-231 cells with transient expression of USP7, USP12, and USP21, respectively, under serial doses of doxorubicin (left panel) or paclitaxel (right panel) treatment, as assed using MTT assays. (**B**) Protein levels of USP7, USP12, and USP21 in MDA-MB-231 cells with transient expression of USP7, USP12, and USP21, respectively, as assessed using western blot analysis. (**C**) Analysis of growth ability of MDA-MB-231 cells with transient expression of USP7 treated with doxorubicin (left panel) or paclitaxel (right panel) using colony formation assays. (**D**) Quantitative analysis of colony formation assays for doxorubicin (left panel) or paclitaxel (right panel)-treated MDA-MB-231-USP7 cells. (**E**) Analysis of apoptosis in MDA-MB-231 with transient expression of USP7, USP12, and USP21, respectively, treated with doxorubicin (left panel) or paclitaxel (right panel) according to Annexin V-dependent flow cytometry assays. (**F**) Quantitative analysis of the apoptotic levels of two different doses of doxorubicin (upper panel)- or paclitaxel (lower panel)-treated MDA-MB-231 cells with transient expression of USP7, USP12, or USP21, respectively. Data are represented as the mean ± SEM for biological triplicate experiments. * *p* < 0.05, compared with the results for control cells.

**Figure 2 cells-11-03294-f002:**
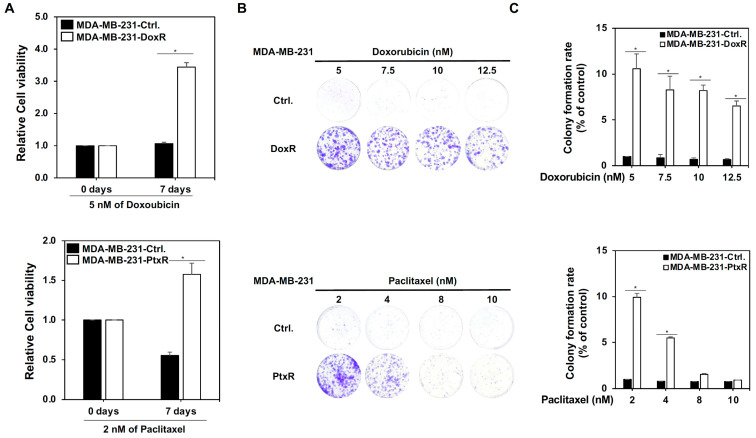
Generation of chemo-drug resistant TNBC. (**A**) Analysis of the viability of doxorubicin-resistant MDA-MB-231 (upper) and paclitaxel-resistant MDA-MB-231 (lower) cells treated with 5 nM of doxorubicin or 2 nM of paclitaxel for 7 days compared with parental MDA-MB-231 cells, assessed using MTT assays. (**B**) Analysis of growth of doxorubicin-resistant MDA-MB-231 (upper) and paclitaxel-resistant MDA-MB-231 (lower) cells compared with parental MDA-MB-231 cells treated with a serial dose of doxorubicin and paclitaxel, assessed performing colony formation assays. (**C**) Quantitative analysis for colony formation assays in MDA-MB-231-DoxR and MDA-MB-231-PtxR cells treated with doxorubicin or paclitaxel, respectively. Data are represented as the mean ± SEM for biological triplicate experiments. * *p* < 0.05, compared with the results for control cells.

**Figure 3 cells-11-03294-f003:**
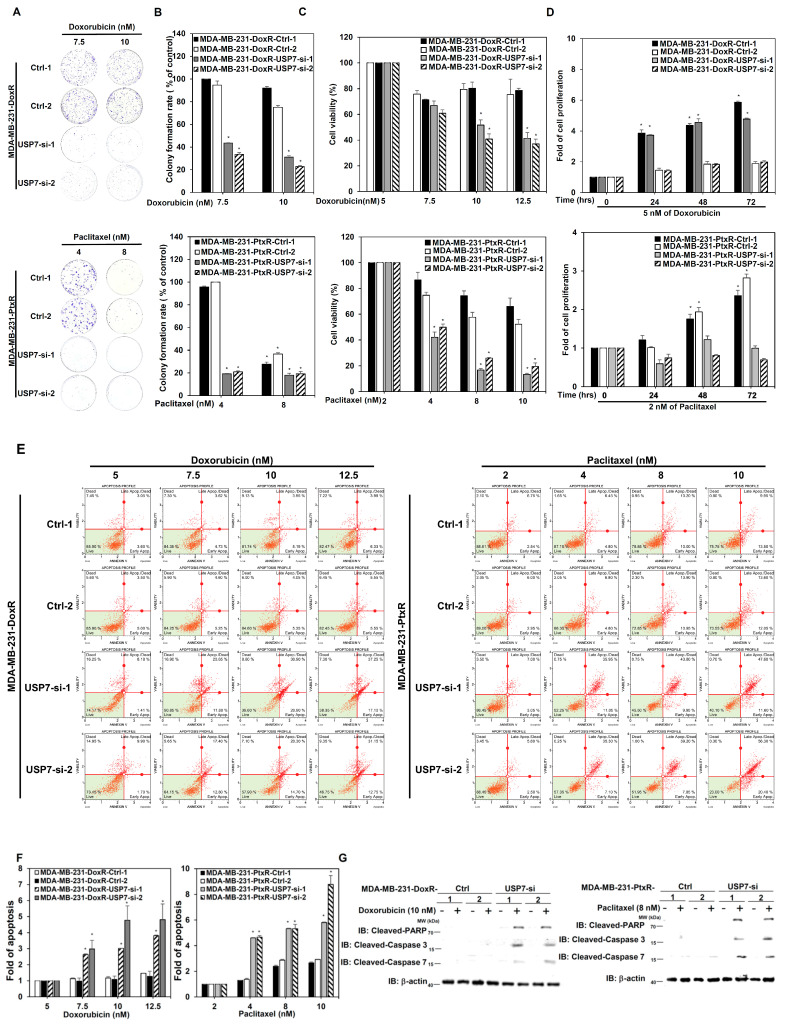
Regulation by USP7 of the chemosensitivity of chemo-drug resistant TNBC. (**A**) Analysis of growth of MDA-MB-231-DoxR-USP7-silencing (upper) and MDA-MB-231-PtxR-USP7-silencing (lower) cells treated with various doses of doxorubicin and paclitaxel, assessed using colony formation assays. (**B**) Quantitative analysis for colony formation assays in MDA-MB-231-DoxR (upper) and MDA-MB-231-PtxR (lower) cells with USP7 suppression under treatment of doxorubicin and paclitaxel, respectively. (**C**) Analysis of the viability in MDA-MB-231-DoxR (upper) and MDA-MB-231-PtxR (lower) cells with USP7 suppression treated with serial doses of doxorubicin and paclitaxel, respectively, performing MTT assays. (**D**) Analysis of the proliferative activity of MDA-MB-231-DoxR (upper) and MDA-MB-231-PtxR (lower) cells with USP7 suppression treated with doxorubicin or paclitaxel for 24, 48, or 72 h, performing MTT assays. (**E**) Analysis of apoptosis in MDA-MB-231-DoxR (left) and MDA-MB-231-PtxR (right) cells with USP7 suppression treated with serial doses of doxorubicin or paclitaxel, assessed by Annexin V-dependent flow cytometry assays. (**F**) Quantitative analysis of apoptotic levels of serial doses of doxorubicin-treated MDA-MB-231-DoxR (left) and paclitaxel-treated MDA-MB-231-PtxR (right) with USP7 suppression. (**G**) Protein levels of cleaved caspase family members in MDA-MB-231-DoxR (left) and MDA-MB-231-PtxR (right) cells with USP7 suppression, treated with serial doses of doxorubicin or paclitaxel, assessed by western blot analysis using specific antibodies. (**H**) Analysis of MMP for MDA-MB-231-DoxR (left) and MDA-MB-231-PtxR (right) cells with USP7 suppression, treated with serial doses of doxorubicin or paclitaxel, assessed using flow cytometry assays. (**I**) Quantitative analysis for total depolarized levels of MMP in MDA-MB-231-DoxR (left) and MDA-MB-231-PtxR (right) with USP7 suppression under treatment of serial doses of doxorubicin and paclitaxel, respectively. (**J**) Bcl family member expression in MDA-MB-231-DoxR (left) and MDA-MB-231-PtxR (right) cells with USP7 suppression under treatment of serial doses of doxorubicin or paclitaxel, by western blot analysis with specific antibodies. Data are represented as the mean ± SEM for biological triplicate experiments. * *p* < 0.05, compared with cells with steady concentration of doxorubicin and paclitaxel, respectively.

**Figure 4 cells-11-03294-f004:**
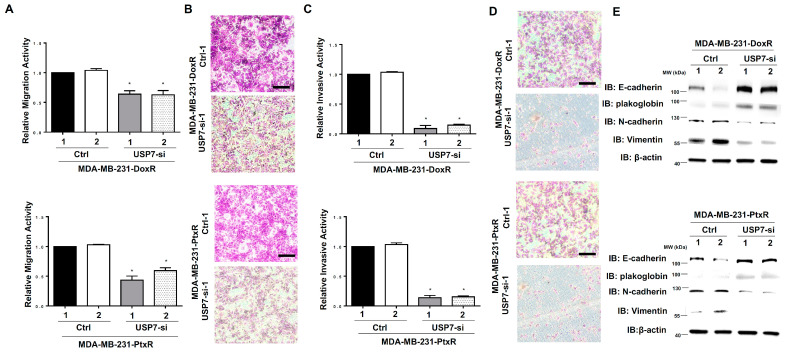
Effect of USP7 suppression on the metastasis of chemoresistant TNBC. (**A**) Relative migration activity of MDA-MB-231-DoxR-USP7-silencing (upper) and MDA-MB-231-PtxR-USP7-silencing (lower) cells, by using in vitro Transwell-dependent migration assays. (**B**) Images for migration activity of MDA-MB-231-DoxR (upper) and MDA-MB-231-PtxR (lower) cells with USP7 suppression with steady concentration of doxorubicin and paclitaxel, respectively. (**C**) Relative invasive activity of MDA-MB-231-DoxR-USP7-silencing (upper) and MDA-MB-231-PtxR-USP7-silencing (lower) cells, using in vitro Transwell-dependent invasive assays. (**D**) Images for the invasive activity of MDA-MB-231-DoxR (upper) and MDA-MB-231-PtxR (lower) cells with USP7 suppression with steady concentration of doxorubicin and paclitaxel, respectively. (**E**) Expression levels of the EMT markers in MDA-MB-231-DoxR-USP7-silencing (upper) and MDA-MB-231-PtxR-USP7-silencing (lower) cells, by western blot analysis with specific antibodies. Data are represented as the mean ± SEM for biological triplicate experiments. * *p* < 0.05, compared with the results for MDA-MB-231-DoxR-Ctrl-1 and MDA-MB-231-PtxR-Ctrl-1 cells, respectively.

**Figure 5 cells-11-03294-f005:**
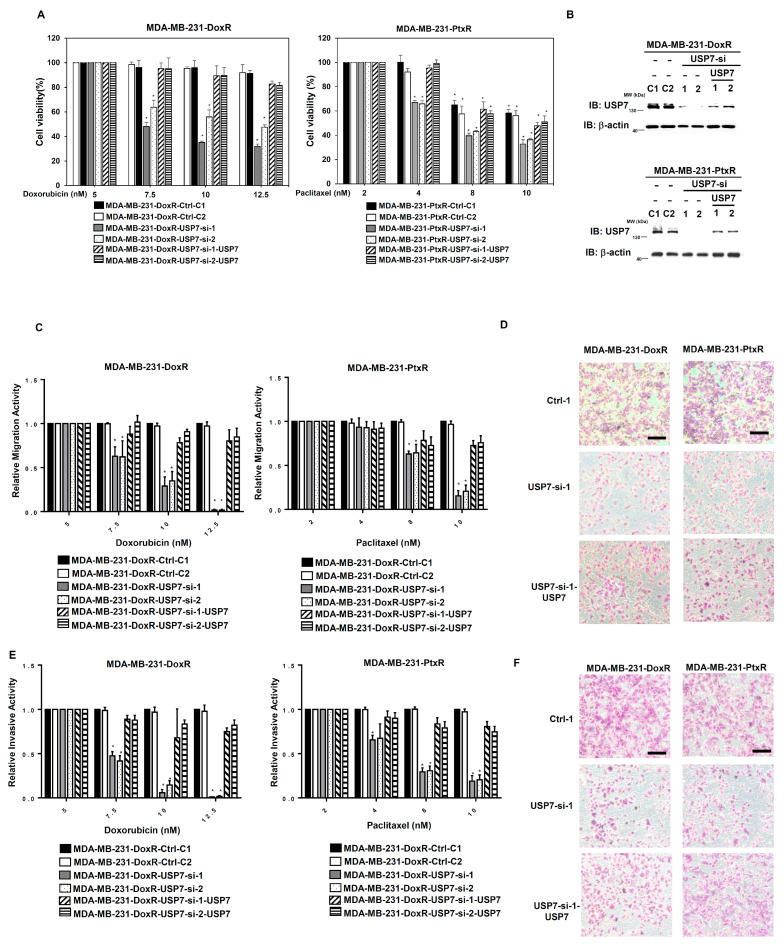
Effect of re-expression of USP7 in chemoresistant triple-negative breast cancer-USP7-silencing cells on cell viability and metastasis. (**A**) Analysis of the viability of MDA-MB-231-DoxR-USP7-silencing (left) and MDA-MB-231-PtxR-USP7-silencing (right) cells with empty vector/USP7 expression treated with serial doses of doxorubicin or paclitaxel, using MTT assays. (**B**) USP7 expressions in MDA-MB-231-DoxR-USP7-silencing (upper) and MDA-MB-231-PtxR-USP7-silencing (lower) cells with empty vector/USP7 expression, by western blot analysis with specific antibodies. (**C**) Relative migration activity of MDA-MB-231-DoxR-USP7-silencing (left) and MDA-MB-231-PtxR-USP7-silencing (right) cells with empty vector/USP7 expression treated with serial doses of doxorubicin and paclitaxel, respectively, using in vitro Transwell-dependent migration assays. (**D**) Images for migration activity of MDA-MB-231-DoxR-USP7-silencing (left) and MDA-MB-231-PtxR-USP7-silencing (right) cells with empty vector/USP7 expression treated with 10 nM of doxorubicin and 8 nM of paclitaxel, respectively. (**E**) Relative invasive activity of MDA-MB-231-DoxR-USP7-silencing (left) and MDA-MB-231-PtxR-USP7-silencing (right) cells with empty vector/USP7 expression treated with serial doses of doxorubicin or paclitaxel, respectively, using in vitro Transwell-dependent invasion assays. (**F**) Images for the invasive activity of MDA-MB-231-DoxR-USP7-silencing (left) and MDA-MB-231-PtxR-USP7-silencing (right) cells with empty vector/USP7 expression treated with 10 nM of doxorubicin or 8 nM of paclitaxel, respectively. Data are represented as the mean ± SEM for biological triplicate experiments. * *p* < 0.05, compared with cells with steady concentration of doxorubicin and paclitaxel, respectively.

**Figure 6 cells-11-03294-f006:**
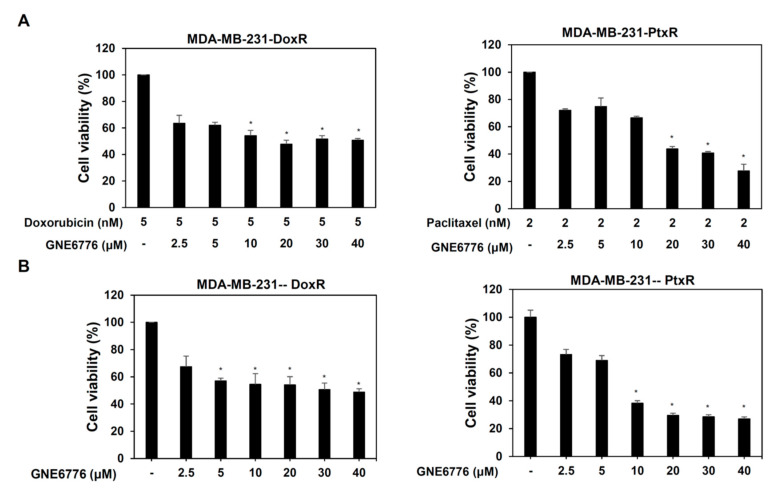
Effect of USP7 inhibitor on cell viability of chemoresistant TNBC. (**A**) The cell viability in MDA-MB-231-DoxR (left) and MDA-MB-231-PtxR (right) cells treated with serial doses of USP7 inhibitor (GNE-6776) combined with doxorubicin or paclitaxel, respectively. (**B**) The cell viability of MDA-MB-231-DoxR (left) and MDA-MB-231-PtxR (right) cells treated with serial doses of USP7 inhibitor (GNE-6776). (**C**) Apoptotic levels in MDA-MB-231-DoxR (upper) and MDA-MB-231-PtxR (lower) cells under serial doses of GNE-6776 treatment, assessed performing Annexin V-dependent flow cytometry assays. (**D**) Analysis for apoptosis of GNE-6776 treated-MDA-MB-231-DoxR (upper) and MDA-MB-231-PtxR (lower) cells. (**E**) Expressions of cleaved caspase family members in MDA-MB-231-DoxR (upper) and MDA-MB-231-PtxR (lower) cells under GNE-6776 treatment, by western blotting analysis with specific antibodies. (**F**) Levels of MMP in MDA-MB-231-DoxR (upper) and MDA-MB-231-PtxR (lower) cells under serial doses of GNE-6776, assessed using flow cytometry assays. (**G**) Analysis of total depolarized levels of MMP in GNE-6776-treated MDA-MB-231-DoxR (upper) and MDA-MB-231-PtxR (lower) cells. (**H**) Expression of Bcl family members in MDA-MB-231-DoxR (upper) and MDA-MB-231-PtxR (lower) cells under serial doses of GNE-6776, by western blot analysis with specific antibodies. Data are represented as the mean ± SEM of biological triplicate experiments. * *p* < 0.05, compared with cells with steady concentration of doxorubicin and paclitaxel, respectively.

**Figure 7 cells-11-03294-f007:**
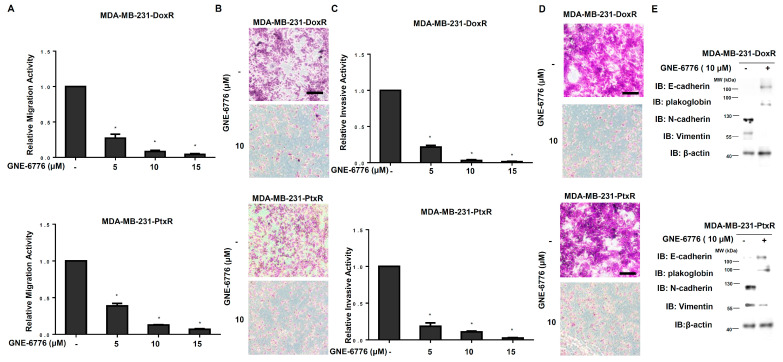
Effect of USP7 inhibitor on metastatic activity of chemoresistant TNBC. (**A**) Relative migration activity in MDA-MB-231-DoxR (upper) and MDA-MB-231-PtxR (lower) cells under serial doses of GNE-6776 by performing in vitro Transwell-dependent migration assays. (**B**) Images for migration activity of GNE-6776-treated MDA-MB-231-DoxR (upper) and MDA-MB-231-PtxR (lower) cells. (**C**) Relative invasive activity of MDA-MB-231-DoxR (upper) and MDA-MB-231-PtxR (lower) cells under serial doses of GNE-6776, using in vitro Transwell-dependent invasive assays. (**D**) Images for the invasive activity of GNE-6776-treated MDA-MB-231-DoxR (upper) and MDA-MB-231-PtxR (lower) cells. (**E**) Protein levels of EMT markers in MDA-MB-231-DoxR (upper) and MDA-MB-231-PtxR (lower) cells under serial doses of GNE-6776, by western blot analysis with specific antibodies. Data are represented as the mean ± SEM for biological triplicate experiments. * *p* < 0.05, compared with cells with steady concentration of doxorubicin and paclitaxel, respectively.

**Figure 8 cells-11-03294-f008:**
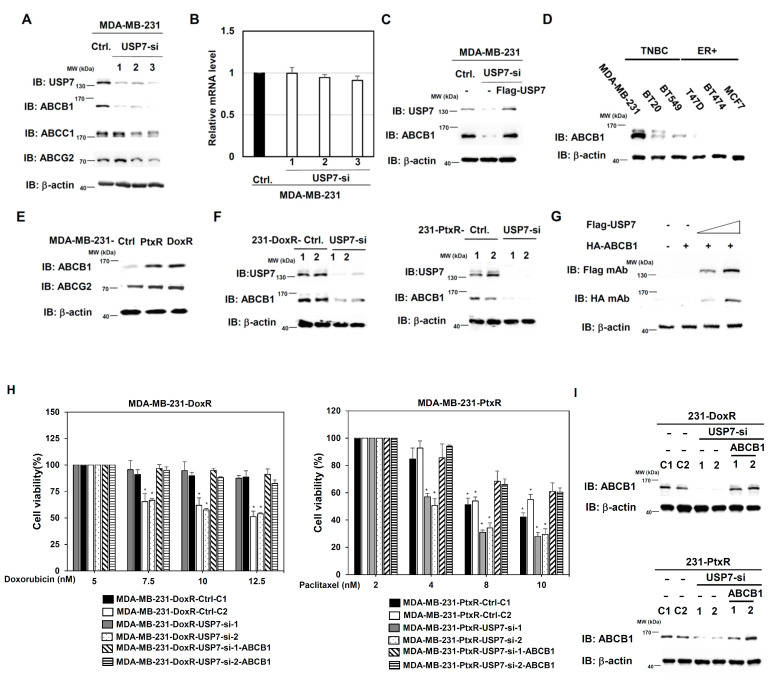
ABCB1 expression is upregulated by USP7 in chemo-drug resistant TNBC. (**A**) Levels of ABC family members in MDA-MB-231 cells with USP7 suppression, by western blot analysis with specific antibodies. (**B**). Analysis of the level of ABCB1 mRNA in MDA-MB-231 cells with USP7-suppression, assessed performing quantitative-RT-PCR analysis. (**C**) ABCB1 levels in MDA-MB-231-USP7-silencing cells with USP7-re-expression, by western blot analysis with specific antibodies. (**D**) Level of ABCB1 in triple-negative and ER-positive breast cell lines, assessed using western blot analysis with specific antibodies. (**E**) Levels of ABCB1 and ABCG2 in MDA-MB-231-DoxR and MDA-MB-231-PtxR cells, assessed using western blot analysis with specific antibodies. (**F**) Levels of ABCB1 in MAD-MB-231-DoxR (left) and MDA-MB-231-PtxR (right) cells with USP7-suppression, assessed using western blot analysis with specific antibodies. (**G**) ABCB1 protein expressions in 293T cell with transient dose-dependent overexpression of USP7, by western blot analysis with specific antibodies. (**H**) Cell viability of MDA-MB-231-DoxR-USP7-silencing (left) and MDA-MB-231-PtxR-USP7-silencing (right) cells with empty vector/ABCB1 expression treated with serial doses of doxorubicin or paclitaxel, assessed performing MTT assays. (**I**) Expression levels of ABCB1 in MDA-MB-231-DoxR-USP7-silencing (upper) and MDA-MB-231-PtxR-USP7-silencing (lower) cells with empty vector/ABCB1 expression, by western blot analysis with specific antibodies. Data are represented as the mean ± SEM for biological triplicate experiments. * *p* < 0.05, compared with cells with steady concentration of doxorubicin and paclitaxel, respectively.

**Figure 9 cells-11-03294-f009:**
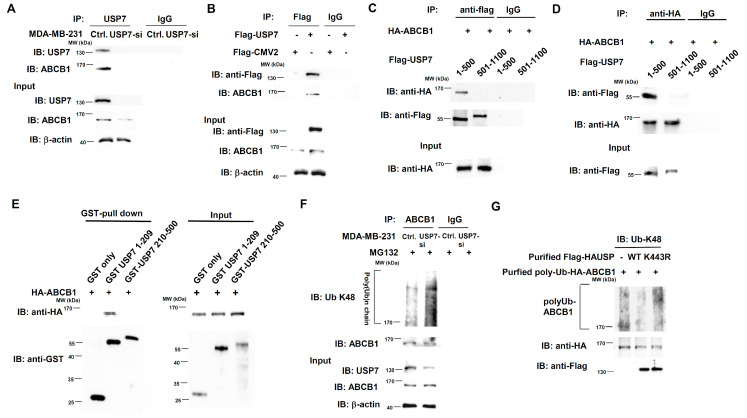
Direct regulation of ABCB1 stability by USP7. (**A**) Interaction between endogenous USP7 and endogenous ABCB1 in MDA-MB-231-USP7-silencing cells, assessed using co-immunoprecipitation assays. (**B**) Interaction between USP7 and ABCB1 in 293T cells with transient overexpression of Flag-tag USP7, assessed using co-immunoprecipitation assays. (**C**) Domain mapping of USP7 interaction with ABCB1, assessed using co-immunoprecipitation assays with anti-flag antibodies. (**D**) Domain mapping of USP7 interaction with ABCB1, assessed using co-immunoprecipitation assays with anti-HA antibodies. (**E**) Direct interaction of USP7 and ABCB1, assessed using GST-pull down assays. (**F**) Levels of K48-linked polyubiquitinated ABCB1 in MDA-MB-231-USP7-silencing cells under MG132 treatment, assessed using in vivo deubiquitination assays. (**G**) Effect of USP7 WT and K443R mutant on K48-linked polyubiquitinated ABCB1, assessed using in vitro deubiquitination assays.

**Figure 10 cells-11-03294-f010:**
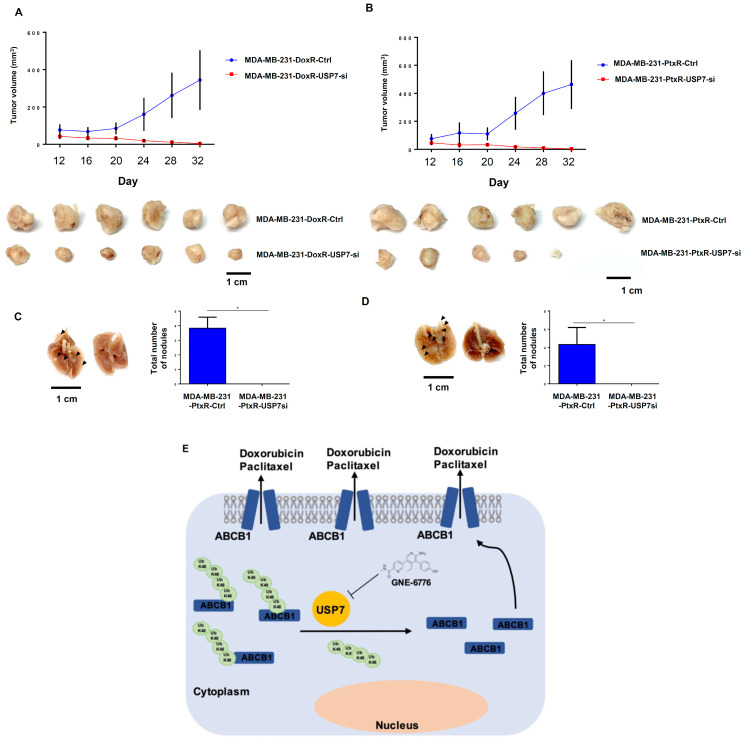
Regulation by USP7 of tumor growth and lung metastasis in chemoresistant TNBC in an orthotopically xenograft animal model. (**A**,**B**) MDA-MB-231-DoxR USP7-silencing and MDA-MB-231-PtxR-USP7-silencing cells were orthotopically injected into Nod SCID mice. Tumor volumes were measured every four days and calculated using the formula V = (L × W^2^)/2. Six mice per group. (**C**,**D**) Picture of a control lung lobe with metastases and a USP7-silencing group lung lobe without metastases. (**E**) A graphic model for the finding in this report. Data are represented as the mean ± SD for animal experiments. * *p <* 0.05, compared with the results for MDA-MB-231-DoxR-ctr1 and MDA-MB-231-PtxR-ctrl cells.

## Data Availability

The data shown in this study is available within the article or Appendix A.

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
