# Peer review of "USP7 Induces Chemoresistance in Triple-Negative Breast Cancer via Deubiquitination and Stabilization of ABCB1"

_cells, 2022, doi:10.3390/cells11203294_

Round 1

Reviewer 1 Report

The article by Yueh-Te Lin et. al. entitled “USP7 induces chemoresistance in triple-negative breast cancer via deubiquitination and stabilization of ABCB1” (as ABCB1 is an innovative substrate of USP7). The authors aim to emphasize that USP7 promotes the chemoresistance of TNBC by stabilizing the ABCB1 protein and the manuscript is well-designed. The authors reported the TNBC's chemoresistance was improved by the overexpression of USP7, however, USP7 knockdown effectively increased the chemosensitivity to the TNBC. The methods were adequately described. The results were clearly presented. The conclusions were reinforced by the results. The strength of the study was previously unreported evidence that both chemotherapeutic drugs (doxorubicin- and paclitaxel) induced chemoresistance that could be reversible when treated with a USP7 inhibitor. The manuscript is suitable for publication after fixing some concerns that should be addressed by the authors.

1. The author must write the aim of the study in the abstract and introduction

2. The quality of the figures is poor and should be replaced

3. Authors may include the future perspective based on their findings

4. Authors can investigate downstream signaling pathways of cell growth/ proliferation (eg. RTK/ RAS/MAP-Kinase, PI3K/Akt etc.,)

5. Authors can validate their study using cell cycle markers and cell proliferation markers

6. Minor typographical errors were found throughout the manuscript and should be amended.

Reviewer 2 Report

In this manuscript, authors utilize cell lines in which MDA-MB-231 BC cells were selected for survival in increasing concentration of doxorubicin (MDA-MB-231-DoxR) or paclitaxel (MDA-MB-231-PtxR), to better understand the mechanism of USP7 in acquired resistance in TNBC. The authors used many results to explore the function of USP7 on TNBS from Figure 1 to Figure 7 in vitro. The structure of manuscript need readjustment. Also, rigor of several experiments needs to be improved.

1.    Before testing for USP7, USP12, and USP21 roles in the chemoresistance of TNBC, the expressions of them in breast cancer cells should be examined.

2.    Analysis of growth ability of MDA-MB-468 cells with transient expression of USP7 treated with doxorubicin or paclitaxel should be examined using colony formation assays.

3.    Analysis of apoptosis in MDA-MB-468 with transient expression of USP7, USP12, and USP21, respectively, treated with doxorubicin or paclitaxel should be examined by flow cytometry.

4.     In Fig.1G, analysis of the viability of MDA-MB-468 cells with transient expression of USP12, and USP21, respectively under serial doses of doxorubicin or paclitaxel treatment should be examined using MTT assays.

5.    layout of Fig.1E, 1F and 1G is uncomfortable. Whether the authors could exchange the layout of Fig.1E, 1F and 1G.

6.    The results of Fig2, Fig3, Fig4, Fig5, Fig6 and Fig7 should be assayed in MDA-MB-468 cells or other breast cancer cells as MDA-MB-231 cells done.

7.    The expressions of ABCB1 in breast cancer cells should be examined.

8.    In Fig. 8B, the authors stated that “The mRNA level of ABCB1 was not changed in MDA-MB-231-USP7-silencing cells compared with control (Figure 8B).”. However, the barplot showed that USP7-si-1 and si-2 had a higher Mrna expression of ABCB1, USP7-si-3 had a lower Mrna expression of ABCB1. Also, the error bar of USP7-si-2 is heavy, the authors should repeat this assay to increase the robustness.

9.    The author had validated USP7 could regulated ABCB1 protein expression. However, in Fig. 9F, the expression of ABCB1 is seem to not changed between USP7-si cells and control cells in input group.

10.  The authors showed that USP7-silencing chemoresistant TNBC exhibits significantly reduced tumorigenesis and lung metastasis in orthotopic BC mouse models. The authors could explore whether the effect of USP7 on TNBC tumorigenesis and metastasis is mediated by ABCB1 in vivo.

11.  The resolution of the apoptosis graph is not high, the authors could improve the resolution of them.

Round 2

Reviewer 2 Report

The reviewer had no other questions.